# Direct band-gap crossover in epitaxial monolayer boron nitride

C. Elias[1], P. Valvin[1], T. Pelini[1], A. Summerfield [2], C.J. Mellor[2], T.S. Cheng[2], L. Eaves[2], C.T. Foxon[2], P.H. Beton [2], S.V. Novikov [2], B. Gil[1] & G. Cassabois[1]

Hexagonal boron nitride is a large band-gap insulating material which complements the electronic and optical properties of graphene and the transition metal dichalcogenides. However, the intrinsic optical properties of monolayer boron nitride remain largely unexplored. In particular, the theoretically expected crossover to a direct-gap in the limit of the single monolayer is presently not confirmed experimentally. Here, in contrast to the technique of exfoliating few-layer 2D hexagonal boron nitride, we exploit the scalable approach of high-temperature molecular beam epitaxy to grow high-quality monolayer boron nitride on graphite substrates. We combine deep-ultraviolet photoluminescence and reflectance spectroscopy with atomic force microscopy to reveal the presence of a direct gap of energy 6.1 eV in the single atomic layers, thus confirming a crossover to direct gap in the monolayer limit.

[1] Laboratoire Charles Coulomb, UMR5221 CNRS-Université de Montpellier, 34095 Montpellier, France. [2] School of Physics and Astronomy, University of Nottingham, Nottingham NG7 2RD, UK. Correspondence and requests for materials should be addressed to G.C. (email: guillaume.cassabois@umontpellier.fr)

The exfoliation of graphene by Novoselov and Geim in 2004[1] opened up a new avenue of research with the advent of graphene, other atomically thin two-dimensional (2D) crystals and related van der Waals heterostructures[2]. In contrast to graphite, transition metal dichalcogenides display a finite energy gap. In exfoliated ultrathin crystals of molybdenum disulphide, Mak et al. demonstrated the transition from indirect band-gap in the bulk crystal to direct gap in a single monolayer[3]. The highly efficient light–matter coupling in a direct-gap monolayer was later observed in other compounds, making the family of semiconducting transition metal dichalcogenides a promising platform for optoelectronics and valleytronics[4].

Hexagonal boron nitride (hBN) is an insulating analogue of graphite with a hexagonal crystal structure and a small lattice mismatch (~1.8%) with graphite, and so is an 'ideal substrate' for graphene and a key building block in van der Waals heterostructures[2]. The growth of high-quality hBN crystals in 2004 also triggered an increasing interest in deep ultraviolet (DUV) applications because of the bright luminescent emission of hBN single crystals[5]. The demonstration of lasing at 215 nm[5] and the operation of field-emitter display-type devices in the DUV[6] make hBN a promising new material for DUV optoelectronics, competing with nitride semiconductors of the AlGaN family that are already widely used for blue and ultraviolet lighting[7].

An isolated monolayer of hexagonal boron nitride (mBN) is predicted theoretically to be a direct-gap semiconductor with a band-gap of around 6 eV[8,9] and with indirect–direct crossover similar to that of molybdenum disulphide[3]. It has been demonstrated that bulk hBN is an indirect-gap semiconductor with a band-gap of 5.95 eV[10–15]. The investigation of the direct-gap properties of mBN is an important issue, not only from a fundamental point of view but also for applications in DUV optoelectronics, with the exciting prospect of it becoming an active layer with highly efficient light–matter coupling in the DUV.

In this context, reliance on exfoliated few-layer samples of hBN may prove to be of limited use. Because the lateral size of the exfoliated flakes is limited to around some tens of micrometres, spatially resolved experiments are required. The difficulty in making photoluminescence (PL) measurements in a microscope operating at wavelengths around 200 nm makes cathodoluminescence the preferred tool to study thin hBN crystals[16–21]. Until now, cathodoluminescence measurements on hBN could only resolve the emission spectrum down to six monolayers[20], leaving unanswered the question of luminescence in mBN, either because of potential intrinsic limitations of cathodoluminescence for atomically thin layers of BN, or because of coupling with the substrate.

We resolve this issue using a strategy, which relies on the scalable growth of mBN and enables macroscopic PL and reflectance measurements in the DUV. Our wafer-scale mBN layers are grown by high-temperature molecular beam epitaxy (MBE) on highly oriented pyrolytic graphite (HOPG) substrates. The use of graphite as the MBE substrate allows us to synthesise mBN by van der Waals epitaxy, with a graphite–mBN interface without adverse reaction or intermixing effects. Whereas reflectance measurements of mBN show a pronounced resonance at around 6.1 eV, PL experiments provide evidence for emission from mBN at this energy. We find that there is no Stokes-shift between reflectance and PL which contrasts with multi-layer and bulk hBN, thus providing a clear signature for a direct-gap in mBN. Furthermore, the DUV cut-off in the optical absorption of graphite makes this material not only a relevant substrate for MBE growth, but also a good one for DUV optoelectronics, as demonstrated by the PL emission of mBN on graphite.

## Results

**Molecular beam epitaxy.** Monolayer BN was first synthesised before the discovery of graphene by decomposition of borazine on a metal surface[22]. The emergence of 2D crystals stimulated various alternative methods of production, such as exfoliation[23] or electron beam thinning[24,25] of hBN films. The production of hBN nanomesh[26] and mBN[27,28] demonstrated the relevance of chemical vapour deposition, culminating with recent report of wafer-scale single-crystal mBN[29]. For the studies reported here, mBN was grown by MBE (see the section "Methods"), a semiconductor technology known to produce atomically flat surfaces and monolayer control of the thickness.

Recently, we reported van der Waals epitaxial growth of hBN on HOPG by high-temperature MBE[30–32]. The hBN layer coverage was controlled by adjusting the epitaxial growth temperature over the range from 1390 to 1690 °C. At the highest growth temperatures, only a small density of hBN islands was observed. The coverage gradually increased to a complete monolayer by decreasing the growth temperature. X-ray photoelectron spectroscopy (XPS) indicates the presence of an interface with no reaction or intermixing effects[33]. Moreover, angle-resolved photoemission spectroscopy (ARPES) shows that the hBN layers are epitaxially aligned with graphite, with a well-defined energy band structure reflecting the high quality of our hBN films[33].

The hBN monolayer coverage can be controlled reproducibly by the MBE growth temperature, time and by the boron:nitrogen flux ratio. The growth of mBN, shown in Fig. 1, was achieved with a growth time of 3 h at a growth temperature $T_g$ of about 1390 °C, and a boron cell temperature $T_B$ of 1875 °C (Supplementary Note 1). Following MBE growth, the hBN layers were characterised using high-resolution tapping-mode (AC-mode) ambient atomic force microscopy (AFM) (see the section "Methods").

**Atomic force microscopy.** For this hBN layer, the HOPG surface is almost completely covered by mBN growth with some 3D hBN aggregates at the graphite step edges, as shown in Fig. 1a and the zoom of a sub-region shown in Fig. 1d. The regions of exposed graphite which remain uncovered by hBN growth are highlighted in the associated phase-channel images (Fig. 1b, e), in agreement with our previous AFM studies of hBN growth on HOPG[30–32]. Thresholding the phase-channel data provides a measurement of the surface coverage of mono-layer and few-layer hBN in addition to the 3D hBN aggregates. This analysis of the data shown in Fig. 1a gives an overall surface coverage of ~95% with ~87% of the surface covered predominantly by mBN together with some small regions of bi-layer, tri-layer and thicker multi-layer hBN near the 3D hBN deposits that nucleate at the HOPG step edges. The monolayer height of the grown mBN layer at the interface with the uncovered HOPG surface is shown in Fig. 1c. The associated line profile (Fig. 1f) shows a step height of ~0.35 nm, as expected for mBN.

**Reflectance.** Reflectance spectroscopy in the DUV (Fig. 2) provides an insight into the optoelectronic properties of these MBE-grown epilayers. We have developed a home-made reflectance setup in order to perform measurements down to 185 nm in a $N_2$-purged atmosphere and a cryogenic environment (see the section "Methods"). Figure 2a displays our experimental results in a spectral domain ranging from 300 to 185 nm, for three different samples at 10 K: graphite (grey line), mBN on graphite (blue line) and bulk hBN (red line). The reflectance spectrum of the bare graphite substrate (grey line) shows a smooth variation with a maximum reflectivity of ~35% around 5.2 eV, as expected from

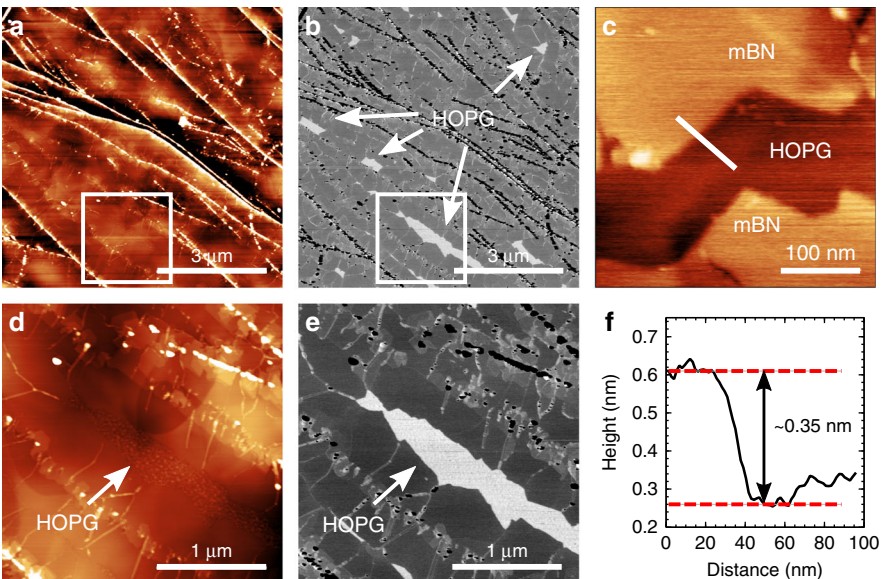

**Fig. 1** Surface morphology of epitaxial monolayer BN on graphite. Atomic force microscopy (AFM) of mBN grown on HOPG (boron cell temperature $T_B =$ 1875 °C, growth temperature $T_g = 1390$ °C and growth time of 3 h). **a** Large-area AC-mode AFM topography of mBN growth on HOPG: the brighter (i.e. topographically higher) regions are due to 3D aggregates of hBN at HOPG step edges. **b** Phase-channel data for image **a**: the white arrows indicate regions of exposed HOPG, uncovered by hBN growth. **c** Small area contact-mode AFM image of the mBN boundary next to an exposed region of HOPG, showing the characteristic monolayer BN step height. **d** Zoom of the region indicated by the white box in **a**. **e** Phase-channel image for image **d**. **f** Line-profile along the interface between mBN and exposed HOPG as indicated by the white line in image **c** showing the characteristic monolayer BN step height

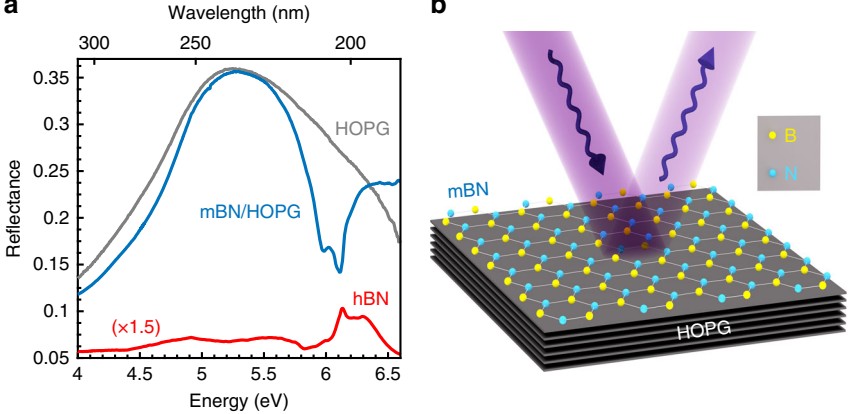

**Fig. 2** Reflectance of epitaxial monolayer BN on graphite. **a** Reflectance spectrum in the deep ultraviolet for the bare graphite substrate (grey line), mBN on graphite (blue line), and bulk hBN (red line), at 10 K. **b** Schematic of the reflectance experiment performed on the mBN-graphite heterostructure synthesised by van der Waals epitaxy. HOPG stands for highly oriented pyrolytic graphite

the DUV dispersion of the refractive index in graphite[34]. The epitaxy of a single monolayer of BN on graphite drastically changes the reflectance spectrum (blue line in Fig. 2a), with a pronounced dip down to 15%-reflectivity at 6.1 eV. Compared to bare graphite, this means a reduction by a factor one-half for the reflectance of mBN on graphite, attesting to the highly efficient light–matter coupling between mBN and graphite. This phenomenology was previously reported in transition metal dichalcogenides, where the ultrastrong optical response in direct-gap monolayers leads to a pronounced extinction of the incident light by the reflected electro-magnetic field[35]. In contrast, for bulk hBN, the indirect nature of the band-gap inhibits light–matter coupling at the energy of 5.95 eV, so that no resonance is observed at 5.95 eV in the reflectance spectrum of bulk hBN (red line in Fig. 2a). The secondary minimum at 6 eV in the reflectance

spectrum of mBN on graphite (blue line in Fig. 2a) is interpreted as a defect-induced brightening of the triplet dark exciton or as trion absorption (Supplementary Note 2).

Before considering the PL emission of mBN, we note also the reflectance spectrum of bare graphite (grey line in Fig. 2a) and in particular its decrease above 5.2 eV. This non-monotonic behaviour results from a key property of graphite in the DUV. The dispersion of the real ($\epsilon_1$) and imaginary ($\epsilon_2$) parts of the dielectric permittivity of graphite were measured in ref. [34] and are plotted in Supplementary Note 1 for the DUV spectral range (Supplementary Fig. 3). Of particular interest is the decrease of $\epsilon_2$ above 4.5 eV, corresponding to a reduction in the optical absorption of graphite in the DUV. Such an effect is crucial for the PL emission of an active layer on top of graphite, since the radiation of a dipole is quenched when it is close to a lossy

dielectric or metallic medium[36,37]. In the context of DUV optoelectronics, graphite appears to be very suitable as an optical substrate for BN. This is a major asset since it is an almost lattice-matched growth template for mBN.

**Photoluminescence.** The optical emission of mBN was probed by PL measurements using our experimental setup optimised for DUV spectroscopy (see the section "Methods"). Figure 3a displays the PL spectrum recorded at 10 K for mBN (blue solid line). We observe a broad band centred at 5.6 eV and sharp lines at 6.05 and 6.08 eV, superimposed on stray light from the laser excitation at higher energy. Only stray light is recorded in the reference spectrum on bare graphite without mBN (grey dotted line). While the broad emission around 5.6 eV was attributed to defects in previous studies on hBN epilayers[31,38], the doublet at higher energy is a unique new feature, which is specific to mBN and resonant with its reflectance minimum (green symbols). The doublet at 6.05 and 6.08 eV is the signature of DUV emission in the direct-gap mBN, as discussed further below. In particular, selective excitation spectroscopy (Fig. 3b) and temperature-dependent measurements (Fig. 3c) illustrate the influence of the atomically thin structure of mBN on its DUV emission.

The PL spectrum of bulk hBN is plotted for comparison in Fig. 3a, as shown by the red solid line. In contrast to transition metal dichalcogenides[3] we note the PL signal intensity of the indirect-gap bulk hBN is more intense than in the direct-gap mBN due to the unusually high internal quantum efficiency in bulk hBN[5,39]. Because of the indirect nature of the band-gap of bulk hBN, the PL signal is red-shifted with respect to the indirect gap at 5.95 eV[10–14]. Consequently, there is no possible overlap of the PL spectrum of bulk hBN with the narrow PL lines of mBN at 6.05 and 6.08 eV, as shown in Fig. 3a. This rules out any misinterpretation of the PL lines at 6.05 and 6.08 eV as being due to carrier recombination in the indirect-gap bulk hBN.

More importantly, this conclusion can also be extended to few-layer hBN crystals because of a curiosity regarding Coulomb screening. Besides the direct–indirect crossover, the mBN–hBN system presents the peculiarity that direct excitons lie at about the same energy in mBN and hBN. Upon reducing the number of layers in bulk hBN, Coulomb interactions are less and less screened, leading to an increase of repulsive electron–electron renormalisation and attractive electron–hole binding. It turns out that the two effects compensate each other[9], so that the direct exciton in mBN is resonant with the direct excitons in hBN, lying ~0.15 eV above the fundamental indirect exciton[40]. In bulk hBN, one thus expects an absorption peak at 6.1 eV, arising from the contribution of direct excitons, in agreement with our reflectance measurements showing a reflectance extremum at 6.1 eV (Fig. 2a, red line). The fact that this extremum is a maximum in bulk hBN, and not a minimum as in mBN, is consistent with the phenomenology of reflectance spectroscopy in bulk crystals[41,42] and will be discussed in detail elsewhere.

We note that in few-layer hBN with an indirect band-gap down to two monolayers[9], phonon-assisted recombination is predicted theoretically to occur at the same energies as in bulk hBN. This is precisely what was observed down to six monolayers for the cathodoluminescence measurements[20]. In that reference, the phonon-assisted recombination lines involving acoustic phonons become dominant as the number of hBN layers is reduced, and they maintain the same energy of 5.9 eV, as in bulk hBN. In the PL spectrum of Fig. 3a (blue line), we also observe a weak emission band at 5.9 eV, arising from the few-layer hBN regions on the surface of our mBN epilayer (Fig. 1).

With increasing thickness of our BN epilayers, the PL lines at 6.05 and 6.08 eV disappear (see Supplementary Note 3), further confirming that they correspond to the DUV emission in the direct-gap mBN. The observation of a doublet PL line rather than a singlet may, at first sight, seem puzzling in direct-gap mBN. However, this is where the valley-physics specific to hexagonal 2D crystals and the existence of momentum-dark intervalley excitons is relevant. As is for the case of graphene and transition metal dichalcogenides, mBN has energy band extrema at the two inequivalent K and K′ points of the Brillouin zone. Four types of exciton can be constructed with an electron and a hole either at K or K′[43]. With both electron and hole in the same valley, one obtains two zero-momentum configurations, corresponding to the expected direct exciton in the monolayer. In addition, there are two excitons with finite centre-of-mass momenta where electron and hole are in different valleys. These indirect excitons in direct-gap compounds are usually called momentum-dark

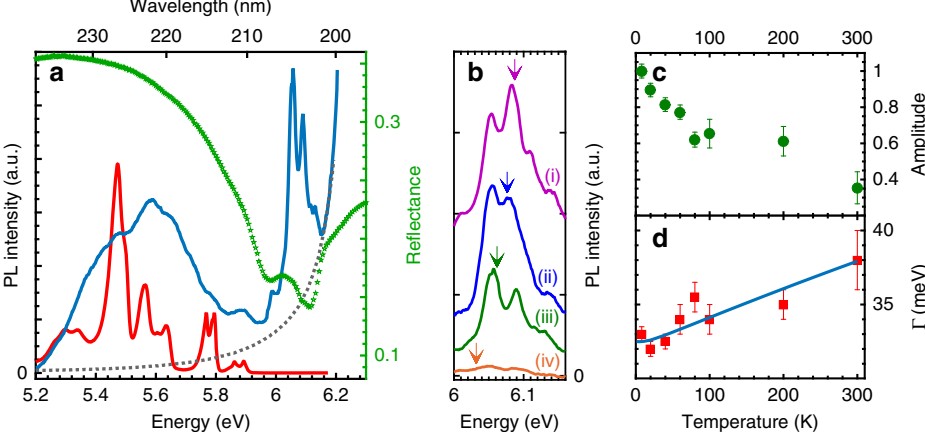

**Fig. 3** Photoluminescence of epitaxial monolayer BN on graphite. **a** Reflectance spectrum (green symbols) and photoluminescence (PL) spectrum (blue solid line) of mBN on graphite at 10 K. Reference emission spectrum recorded on bare graphite without mBN (grey dotted line) showing stray light from the laser excitation. The normalised PL spectrum in bulk hBN at 10 K is plotted as a red line for comparison. **b** Emission of mBN (after subtraction of laser stray light) for different values of the excitation energy $E_{ex}$: (i) 6.4, (ii) 6.39, (iii) 6.375, and (iv) 6.345 eV from top to bottom (the three top spectra are shifted vertically for clarity). The vertical arrows indicate the Raman-shifted energy $E_{ex}-2\Delta$, with $\Delta = 156$ meV, corresponding to the energy of the LA(M) phonon in mBN. The vertical arrows point at: (i) 6.088, (ii) 6.078, (iii) 6.063, and (iv) 6.033 eV. **c** Normalised amplitude of the PL doublet lines versus temperature. **d** Full width at half maximum Γ of the lines in the PL doublet from 10 K to room temperature: data (symbols), fit (solid line). Error bars indicate the standard deviations for least-squares fitting of the PL spectra

excitons, and their recombination requires the emission of a phonon at the K point of the Brillouin zone. High-quality MoSe$_2$ monolayers with narrow emission lines have recently resolved the intrinsic structure of the PL spectrum[44]. The PL spectrum consists of a doublet, closely analogous to those observed in our mBN (Fig. 3a). As for the case in ref. [44], we interpret the PL line at 6.08 eV as the direct recombination of an exciton with both electron and hole in the same valley, while the line at 6.05 eV arises from phonon-assisted recombination of the momentum-dark exciton in mBN with the emission of an out-of-plane phonon. This splitting of 34 meV in the PL doublet (see Supplementary Note 5 for details on line shape fitting) matches the energy of the out-of-plane acoustic phonon ZA(K) in mBN[45,46]. Finally, we note the complexity of the PL spectrum goes beyond a simple doublet with additional peaks of lower intensity than the dominant doublet. While the 5.98 eV component may arise from the brightened triplet or the trion (see Supplementary Note 2), the identification of the weak shoulders on the high-energy side will require more experimental and theoretical work in order to obtain a comprehensive understanding of the full emission spectrum.

In addition to the indirect–direct crossover in mBN, the exciton-binding energy is also strongly enhanced compared to the bulk case due to the reduced screening of Coulomb interactions in monolayers. In free-standing mBN, calculations predict an exciton-binding energy of 1.9 eV compared to 0.7 eV for bulk hBN[9], i.e. a 'giant' exciton-binding energy in mBN and the highest value among 2D crystals. Although the graphite substrate enhances screening in our epitaxial mBN, our measurements indicate that there are no possible excitations of excited excitonic states or free electron–hole pairs in our mBN, in contrast to transition metal dichalcogenides[4]. This implies that the giant exciton-binding energy greatly modifies optical pumping in mBN, as we will now explain.

In hBN, PL spectroscopy is routinely performed with DUV excitation around 195 nm, either provided by excimer lasers or by the fourth-harmonic signal generated from a Ti:Sa laser. These laser sources enable the generation of electron–hole pairs in the absorption continuum of bulk hBN. In contrast, for mBN, due to the giant exciton-binding energy, the first excited state is located 0.9 eV above the fundamental exciton[9], i.e. at 7 eV (177 nm), corresponding to a spectral window beyond the reach of standard DUV laser sources.

In order to circumvent this critical issue, we used resonant excitation of phonon modes. In this scheme, optical pumping is performed with a detuning between excitation and detection given by the energy of one or more phonons. This method of optical pumping is the key for detecting the PL signal in mBN, as shown in Fig. 3b. In this figure, the PL spectra around 6.08 eV for different values of the excitation energy $E_{ex}$ are presented. Strong modifications of the PL signal intensity are observed. In particular, there is barely any emission for $E_{ex} = 6.345$ eV, whereas an intense PL signal would be recorded in bulk hBN for this excitation energy. An efficient excitation pathway is obtained for $E_{ex}$ ranging from 6.375 to 6.4 eV, i.e. ~300 meV above the emission energy of mBN. We also note a distortion of the emission spectrum when varying $E_{ex}$, arising from the intrusion of resonant Raman scattering, 312 meV below the excitation energy (vertical arrows in Fig. 3b), and superimposed upon the PL doublet (for more details, see Supplementary Note 4).

For monolayer MoSe$_2$, PL excitation spectroscopy has provided evidence for the prominent effect of longitudinal acoustic phonon modes at the M point, labelled LA(M), in the resonant excitation of phonon modes[47]. This result is consistent with ab initio calculations in MoS$_2$ and WS$_2$, predicting the strongest

electron–phonon interaction for LA phonons in the vicinity of the M points[47]. In mBN, the LA(M) energy is calculated to be at $150 \pm 5$ meV[45,46]. In fact, the 2LA(M) overtone has an energy of $300 \pm 10$ meV, in excellent agreement with the detuning of 312 meV of the resonant Raman scattering signal with the excitation laser (vertical arrows in Fig. 3b). We therefore conclude that the resonant excitation of the 2LA(M) overtone is an efficient excitation pathway, which provides optical pumping at $6.38 \pm 0.03$ eV ($\lambda = 194.5 \pm 1$ nm) in mBN.

The giant exciton-binding energy and atomically thin structure in mBN has a significant effect on the temperature-dependence of the DUV emission in mBN (Fig. 3c, d). From the quantitative analysis of our data (Supplementary Note 5), we can evaluate the amplitude and linewidth of the PL lines. From 10 K to room temperature, the linewidth $\Gamma$ increases from 32 to 38 meV (Fig. 3d). The 6 meV-thermal broadening in mBN is much smaller than in bulk hBN, where phonon-assisted broadening is predominantly due to the interlayer breathing mode[48]. Such a mode no longer exists in atomically thin mBN, so that the thermal broadening in mBN is limited to quasi-elastic acoustic phonon scattering. The 6 meV-thermal broadening in mBN can be reproduced (solid line, Fig. 3d) by taking the contribution of acoustic phonon scattering in bulk hBN, but corrected by a factor 1.7 (see Supplementary Note 5). Finally, note that the spectrally integrated amplitude of the PL doublet lines decreases only by a factor 3 from 10 to 300 K in mBN (Fig. 3c). This temperature-dependence is weaker than in bulk hBN[14] as a consequence of the very large excitonic-binding energy in mBN. It suggests a high internal quantum efficiency at 300 K, of primary importance for applications of mBN in DUV optoelectronic devices operating in ambient conditions.

## Discussion

Lastly, we emphasise that our demonstration of the direct band-gap crossover in the monolayer limit did not follow the standard methods documented in the literature. Inspections of the absorption edge and of the quantum yield are common tools in semiconductor physics. Surprisingly, they are irrelevant in the present case because bulk hBN is an indirect band-gap semiconductor with optoelectronic properties strikingly resembling a direct-gap semiconductor. First, the absorption spectrum does not show broad bands but narrow lines and sharp edges[10]. It arises from the peculiar band structure of hBN. It leads to the involvement of phonons with a finite group velocity in the recombination processes, which itself results in a strong energy-dependence of the exciton-phonon matrix element[10]. Thus, inspection of absorption edges in indirect-gap bulk hBN vs. direct-gap mBN is irrelevant to the demonstration of the direct band-gap crossover. Second, the same holds for the quantum yield. Indirect-gap bulk hBN is a bright emitter, very much like direct gap semiconductors. This specificity was first pointed out in the pioneering work of Watanabe and Taniguchi on high-quality hBN crystals[5] and it led to the misinterpretation of bulk hBN as a direct-gap material. Recently, this qualitative description was complemented by quantitative estimations of the quantum yield, of the order of 40–50% in bulk hBN, a remarkably large value for an indirect-gap material[39]. Therefore, comparison of the quantum yield in indirect-gap bulk hBN vs. direct-gap mBN is also not appropriate for the demonstration of the direct band-gap crossover.

In conclusion, we have used the scalable growth approach of MBE to fabricate high-quality monolayer boron nitride on graphite. Reflectance and PL experiments in the DUV demonstrate the existence of a direct-gap at around 6.1 eV (203 nm) in wafer-scale monolayer boron nitride. The giant exciton-binding energy

in mBN requires a selective optical pumping through resonant excitation of phonons. It also provides a high internal quantum efficiency at room temperature. We emphasise the relevance of the graphite substrate, not only as a nearly lattice-matched growth template for van der Waals epitaxy, but also as a low-loss optical substrate for DUV applications, with potential for the scalable development of graphene/boron nitride van der Waals heterostructures and for DUV optoelectronics.

## Methods

**Samples**. The mBN was grown by MBE on graphite substrates. The bulk hBN is a commercial crystal from HQ Graphene (http://www.hqgraphene.com/).

**Molecular beam epitaxy**. BN epilayers were grown using a custom-designed Veeco GENxplor MBE system capable of achieving growth temperatures as high as 1850 °C under ultra-high vacuum conditions, on rotating substrates with diameters of up to 3in. Details of the MBE system can be found in refs. [30–32]. In all our studies we relied on thermocouple readings to measure the growth temperature of the substrate. We used a Veeco high-temperature effusion cell for evaporation of boron, and a standard Veeco RF plasma source to provide the active nitrogen flux. High-purity (5 N) elemental boron contains the natural mixture of $^{11}B$ and $^{10}B$ isotopes. The BN epilayers were grown using a fixed RF power of 550 W and a nitrogen ($N_2$) flow rate of 2 sccm. We used $10 \times 10$ mm$^2$ HOPG substrates with a mosaic spread of 0.4. The HOPG substrates were prepared by exfoliation using adhesive tape to obtain a fresh graphite surface for epitaxy. After exfoliation, the HOPG substrates were additionally cleaned in toluene overnight to remove any remaining tape residue and annealed in a barrel furnace at 200 °C in a $H_2$:Ar (5%:95%) gas flow for 4 h as a final cleaning step [30–32].

**Atomic force microscopy**. AFM measurements were performed in ambient conditions on the as-grown hBN epilayers on HOPG samples using an Asylum Research Cypher-S system. Samples were imaged using Multi75Ai-G cantilevers (Budget Sensors, supplied by Windsor Scientific. Resonant frequency = 70 kHz) in amplitude-modulated repulsive tapping-mode (AC-mode) at amplitude set-points of 60–70% of the free air amplitude when driven at 5% below resonance. AFM data was processed using the Gwyddion software package [49].

**DUV reflectance and PL**. In our experimental setup, the sample is held on the cold finger of a closed-cycle cryostat for temperature-dependent measurements from 10 K to room temperature.

A home-made reflectance setup was built based on a deuterium lamp (63163 Newport), spectrally filtered through a monochromator (Cornerstone CS130) with 100 µm-slits providing a spectral resolution of order 10 meV. The incident light was focused on the sample close to normal incidence, with a rectangular spot of size 100 µm × 1.5 mm. All optical elements are reflective mirrors coated for DUV. In order to remove absorption lines of $O_2$ in the DUV, the whole optical path was placed in a $N_2$-purged atmosphere.

In PL spectroscopy, the excitation beam was provided either by the fourth or the second harmonic of a cw mode-locked Ti:Sa oscillator with a repetition of 82 MHz, for measurements in mBN and bulk hBN, respectively. The spot diameter is of the order of 50 µm, with a power of 30 µW and 50 mW for the fourth and second harmonic, respectively. An achromatic optical system couples the emitted signal to our detection system, composed of a $f = 300$ mm Czerny–Turner monochromator, equipped with a 1800 grooves mm$^{-1}$ grating blazed at 250 nm, and with a back-illuminated CCD camera (Andor Newton 920), with a quantum efficiency of 50% at 210 nm, operated over integration times of 5 min.

## Data availability

The raw data for the AFM images, and the photoluminescence and reflectance spectra may be accessed through the University of Nottingham Research Data Management Repository at https://doi.org/10.17639/nott.6996.

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

## Acknowledgements

We gratefully acknowledge C. L'Henoret for his technical support at the mechanics workshop, A. Dréau, V. Jacques and I. Philip for fruitful discussions. This work and the Ph.D. funding of C.E. and T.P. were financially supported by the network GaNeX (ANR-11-LABX-0014). GaNeX belongs to the publicly funded *Investissements d'Avenir* programme managed by the French ANR agency. This work was also supported by the Engineering and Physical Sciences Research Council [grant number EP/K040243/1, EP/L013908/1, EP/M013294/1, EP/M50810X/1, EP/P019080/1]; and Leverhulme Trust [Grant No. RPG-2014-129]. We also thank the University of Nottingham Propulsion Futures Beacon for funding towards this research.

## Author contributions

The samples were grown by T.S.C. with additional input on the MBE process from C.T.F. and S.V.N. The HOPG substrates were prepared by A.S.; A.S. and P.H.B. acquired the AFM images, which were interpreted by A.S., P.H.B., C.J.M. and L.E. For optical studies, C.E., P.V., T.P., B.G. and G.C. performed photoluminescence and reflectance spectroscopy. All authors conceived this project and contributed to discussions and interpretation of the results. The manuscript was written by G.C. with inputs from the other authors.

## Additional information

**Competing interests:** The authors declare no competing interests.

