## [Peer Review File · Nature Communications]

Reviewers' comments:

Reviewer #1 (Remarks to the Author):

The manuscript by Elias et al reports optical studies of monolayer hBN (mBN). The combination of experimental developments and samples with extended mBN flakes on HOPG enabled optical studies (reflectance and PL spectroscopy) with sensitivities down to one hBN layer, which in turn allowed the authors to observe the cross-over from an indirect to a direct band-gap semiconductor in the limit of the last layer.

The work is an experimental masterpiece, and it addresses a long-standing question with respect to one of the most fundamental properties of an important material. The manuscript is well written, and the interpretation is to the most extend sound. Beyond any doubt, the work deserves publication in Nature Communications.

However, I find the interpretation of the peaks premature. The authors attribute the lower energy peak in reflectance (also visible in PL) with a 130 meV redshift from the main absorption peak to a defect-brightened optically dark triplet, whereas the two lines of the dominant PL doublet with a 34 meV splitting are assigned to bright exciton and phonon-assisted momentum-dark exciton decay pathways. However, both the reflectance and the PL spectra in Fig. 3 exhibit more fine structure than captured by this model (note e.g. the peak multiplicity in the PL spectra of Fig. 3a and 3b and an additional peak in PL at ~ 6.12 eV that coincides with a dip in reflectance in Fig. 3a; moreover, the dominant reflectance dip seems to consist of two features at ~ 6.08 and 6.12 eV and not just one). Overall, the spectra are complex, and their closed interpretation will require more experimental and theoretical work beyond the scope of the present manuscript. Therefore, the authors could be more conservative in their interpretation and discuss other possible scenarios of assignment as well. As an example, one could assume the ratio of trion to exciton binding energies in mBN to be similar to that in monolayer transition metal dichalcogenides. Within this framework, and consistent with data in Fig. S4, the redshifted peak in reflectance would correspond to trion absorption, and varying sample growth conditions would imply different visibilities of the trion peak due to different levels of growth-induced residual doping. There are certainly other scenarios consistent with the presented experimental data, and I don't see how one of them can be singled out at this point. Therefore, I believe that the manuscript could actually profit from a less explicit interpretation.

Minor points:

- Did the authors attempt the same type of experiments on exfoliated mBN? The spot size of 50 microns in diameter seems to be consistent with the typical extend of exfoliated flakes.

- In the caption of Fig. 3b, the arrows are numbered from (i) to (iv) without any correspondence in the figure

Reviewer #2 (Remarks to the Author):

The manuscript presents an experimental study of reflectance and photoluminescence in hexagonal Boron-Nitride (hBN). Being a material with very wide bandgap, its optical properties exhibit features in the deep ultra-violet (DUV) range, which makes it specially important, e.g. by being somewhat far from the range of luminescence of other similar two-dimensional semiconductors, such as TMDs, phosphorene, etc.

The paper is well written and the conclusions are based in convincing experimental observations.

First studies of DUV photoluminescence in bulk hBN date from more than a decade ago. The topic, in general, is not entirely new. Nevertheless, the paper presents results of significant importance for the development of future opto-electronics in atomically thin BN in the DUV range. A comparison is made between results obtained for this material in its bulk and monolayer (mBN) form. The observation of a PL signal at ~ 6.1 eV in the monolayer case, which is absent in all other samples with higher thickness, sounds like good experimental evidence of the expected indirect-to-direct gap transition from few-layer to monolayer BN. The paper also defends the use of highly oriented pyrolytic graphite (HPOG) as a very good substrate for mBN, not only because it is almost perfectly matched to its crystal structure, but also because HPOG exhibits a convenient decrease in reflectance just before the energy range where mBN features start to appear.

In fact, observing PL in mBN is challenging: due to the lack of screening above the layer, the quasi-particle gap is even deeper in the ultra-violet. Excitation energies would need to be higher in order to pump excitons in the continuum, as compared to those required for PL in bulk hBN. The authors circumvent this problem by using a phonon mediated process. The stronger binding energy is verified by the (relatively weak) temperature dependence of the PL peaks. Even so, I think the authors should mention that the difference of 0.9 eV between the ground and first excited states of mBN in Ref. [9] is calculated assuming a free-standing layer. The presence of graphite in the actual sample presented here will probably enhance screening and thus reduce this difference and binding energies in general. It will most likely be still high, so, conclusions would remain the same.

In summary, I believe the paper is scientifically sound and brings enough novel information about a topic that has been of interest for the past few years. Therefore, I recommend its publication in Nature Communications in its present form.

Reviewer #3 (Remarks to the Author):

The authors have reported the growth of monolayer BN on graphite, the characterization of the reflectance and PL spectra. The dependence of the PL spectra on excitation wavelengths has also been investigated. Based on these results, the authors concluded that they have observed a crossover from indirect band gap few-layer BN to direct band gap in monolayer BN. The authors also reported that excitonic effect is significantly enhanced in the monolayer material compared to the bulk, and resonant excitations through phonons are required to excite these excitons.

After reading through the manuscript, I do not find concrete experimental evidence supporting the claim of a crossover from indirect gap to direct gap in monolayer BN. First, I am not completely convinced based on a line cut in Fig. 1c one could conclude that the sample is mainly consisted of monolayer BN. Second, what is the exact experimental evidence of a crossover to direct gap? I do not see a significant enhancement of PL quantum yield as the layer number decreases, as in the case for transition metal dichalcogenides. I also do not see careful measurement of the optical conductivity showing the indirect gap transition blueshifts to the higher energy side of the direct gap transition. Based on the reflectance and PL spectra, I do not see clear evidence of crossover to direct gap. The enhanced reflectance peak for thin BN on graphite compared to that of bulk BN does not support the crossover. The local electric field is so different in the two cases that the authors need to do a better job to extract the optical conductivity. Since the exciton binding energy is so much bigger than room temperature for both bulk and monolayer BN, I do not see how the exciton oscillator strength can be that different between monolayer and bulk BN. I also do not see evidence of a crossover in the PL studies. The lack of a significant Stoke shift in the grown sample compared to the bulk sample does not support the crossover. In fact, it appears that the PL is much weaker in the monolayer BN, which disagrees with a crossover.

Lastly, it is also unclear to me what I am supposed to see with the arrows in Fig. 3b. The authors claim that they are important in supporting the picture of resonant phonon excitations. But I do not see any features there. Also, there are too few excitation wavelengths performed to support the picture of resonant phonon excitations. Much more careful studies are needed to support the claim. Overall, I do not find the study presents convincing evidence to support the claims. Most of the assignments have been arbitrary in my opinion. I therefore cannot recommend its publication in Nature Communications.

Reviewer #1

Main Point

Besides the positive assessment of our submission, this reviewer believes "*the manuscript could actually profit from a less explicit interpretation*" of the low energy peak in reflectance and dominant PL doublet.

We agree with Reviewer 1 that there are alternative interpretations of these peaks, and we appreciate the constructive comment and suggestion to introduce other possible scenarios involving trions. We have significantly modified our manuscript accordingly (see Changes C1 and C2 below). Change C1 deals with the interpretation of the low energy peak in reflectance: we have modified the corresponding sentence in the main text and incorporated a new paragraph in the Supplementary Material. Change C2 corresponds to the interpretation of the dominant PL doublet: we have commented the peak multiplicity in the PL spectrum and discussed other possible scenarios in addition to the intervalley scattering and the momentum-dark excitons.

Minor point 1 about "*same experiments on exfoliated mBN*"

We have searched for the PL signal from exfoliated mBN deposited on different substrates without any success so far. We suspect the interaction with the substrate plays a detrimental role which is suppressed in the present case due to the special nature of the van der Waals epitaxy on graphite. Moreover, the exfoliation and deposition process can generate surface contaminations which are critical because of the monolayer thickness of mBN. Furthermore, it is possible that organic residues act as low-energy non-radiative centers in the exfoliated mBN.

Minor point 2 about labelling error in Fig.3b

We thank Reviewer 1 for having spotted this mistake and our omission of a proper labelling in Fig.3b. The (i) to (iv) labels have been added in the revised version (see Change C3).

Reviewer #2

Suggestion about "*screening enhancement due to graphite*"

We agree with Reviewer 2 about the influence of graphite on increasing the screening of Coulomb interactions in mBN. We appreciate this constructive comment and we have updated our discussion about the giant exciton binding energy in mBN by noting the possible modification of the screening by the graphite substrate and the corresponding decrease of the splitting of excitonic states (see Change C4).

Reviewer #3

Point 1 about "*sample mainly consisted of monolayer BN*"

Reviewer 3 writes "*I am not completely convinced based on a line cut in Fig. 1c one could conclude that the sample is mainly consisted of monolayer BN*".

Such a statement suggests that Reviewer 3 has disregarded our discussion of the statistical analysis of surface coverage, namely (page 5):

"Thresholding the phase-channel data provides a measurement of the surface coverage of mono- and few-layer hBN in addition to the 3D hBN aggregates. This analysis of the data shown in Fig.1(a) gives an overall surface coverage of ~95% with ~87% of the surface covered predominantly by mBN together with some small regions of bi-, tri- and thicker multi-layer hBN near the 3D hBN deposits that nucleate at the HOPG step edges. The monolayer height of the grown mBN layer at the interface with the uncovered HOPG surface is shown in Fig.1(c). The associated line profile [Fig.1(f)] shows a step height of ~0.35 nm, as expected for mBN."

As explained in the paragraph above, the line cut displayed in Fig.1c&f is an illustration that the ~87% of BN covering the surface are predominantly composed of a monolayer of hBN. By itself, the line cut cannot prove that the surface is mainly covered by mBN. This is the reason why we have performed a statistical analysis of the AFM measurements.

A further detailed discussion about the thicknesses of mBN and other hBN samples is presented in the section "*A1. Growth*" of the Supplementary Material.

Point 2 about "*no significant increase of PL quantum yield*"

Reviewer 3 raises an argument that is particular to studies of transition metal dichalcogenides when writing "*I do not see a significant enhancement of PL quantum yield as the layer number decreases, as in the case for transition metal dichalcogenides.*"

This argument is based on the usual observation of an increase of the PL signal when decreasing the number of layers, with a huge enhancement when switching from two to one monolayers [as shown in the pioneering paper by Mak *et al.* PRL **105**, 136805 (2010)].

This argument cannot hold in hBN because of the unusually high internal quantum efficiency in bulk hBN. This was first pointed out by Watanabe *et al.* in their seminal paper on high-quality hBN crystals [Nat. Mater. **3**, 404 (2004)], and recently quantitatively estimated by Schué *et al.* [PRL **122**, 067401 (2019)]. In the latter reference, it was demonstrated the internal quantum efficiency is as high as 40-50%, a value similar to direct bandgap semiconductors (e.g. ZnO in PRL **122**, 067401 (2019)). Such a phenomenon arises from the strong electron-phonon coupling in hBN, leading to efficient phonon-assisted recombination which is fast enough to bypass non-radiative recombination in hBN. This unusual effect is not yet understood from a theoretical point of view, but experimental facts are there, demonstrating that bulk hBN is a very promising active material for deep UV optoelectronics despite its indirect bandgap.

As a consequence of the high internal quantum efficiency in the indirect-gap bulk hBN, the reduction of the number of layers down to a single monolayer can only lead to a decrease of the PL signal. This explains why "*the PL is much weaker in the monolayer BN*" as written by Reviewer 3.

This explanation suggests that Reviewer 3 is not familiar with the unusually high internal quantum efficiency in bulk hBN. For the sake of clarity, we have added a sentence in order to avoid any misunderstanding of our results in consideration of the PL signal intensity in bulk hBN and mBN (see Change C5).

Point 3 about "*indirect gap transition blueshifting to the higher energy side of the direct gap*"

Reviewer 3 raises another argument coming from the literature on transition metal dichalcogenides: "*I do not see careful measurement of the optical conductivity showing the indirect gap transition blueshifts to the higher energy side of the direct gap transition.*"

In multi-layer transition metal dichalcogenides samples, the broad PL band at low energy is attributed to the fundamental indirect recombination with a blue-shift when decreasing the

number of layers [Mak et al. PRL **105**, 136805 (2010)].

Point 3 of Reviewer 3 is very specific to transition metal dichalcogenides, and Reviewer 3 appears to have disregarded our detailed discussion of the energy of direct and indirect transitions in hBN. On page 7 of our paper, we first comment the absence of variations of the direct exciton energy as a function of the number of layers because of the exact compensation of repulsive electron-electron renormalisation and attractive electron-hole binding. Following this discussion, we also explain that the same holds for the energy of the indirect transition down to two monolayers, namely (page 7):

"We note that in few-layer hBN with an indirect band-gap down to two monolayers [9], phonon-assisted recombination is predicted theoretically to occur at the same energies as in bulk hBN. This is precisely what was observed down to six monolayers for the cathodoluminescence measurements [20]. In that reference, the phonon-assisted recombination lines involving acoustic phonons become dominant as the number of hBN layers is reduced, and they maintain the same energy of 5.9 eV, as in bulk hBN. In the PL spectrum of Fig.3(a) (blue line), we also observe a weak emission band at 5.9 eV, arising from the few-layer hBN regions on the surface of our mBN epilayer [Fig.1]."

We believe that this refutes point 3 of Reviewer 3.

Point 4 about *"the enhanced reflectance peak for thin BN on graphite"*

Reviewer 3 raises the question of the contrast in reflectance measurements, and more specifically the enhanced reflectance peak in thin BN on graphite: *"The enhanced reflectance peak for thin BN on graphite compared to that of bulk BN does not support the crossover... Since the exciton binding energy is so much bigger than room temperature for both bulk and monolayer BN, I do not see how the exciton oscillator strength can be that different between monolayer and bulk BN."*

The answer relates to the different optical configurations: (i) mBN on graphite is a multi-layer system, (ii) bulk hBN is a semi-infinite medium.

In the case of mBN on graphite, the relevant question is the modification of the substrate reflectivity by the atomically-thin mBN. In the text, we wrote (page 5) *"Compared to bare graphite, this means a reduction by a factor one-half for the reflectance of mBN on graphite, attesting to the highly efficient light-matter coupling between mBN and graphite. This phenomenology was previously reported in transition metal dichalcogenides, where the ultrastrong optical response in direct-gap monolayers leads to a pronounced extinction of the incident light by*

the reflected electro-magnetic field [35]." In Ref.[35], the authors demonstrated that the giant extinction of the incident field is a peculiar feature of atomically-thin 2D material leading, in the present case of mBN on graphite, to the strong decrease of the reflected signal compared to bare graphite reported in Fig.2a.

In the case of bulk hBN, we encounter the standard problem of reflectivity at the interface of a semi-infinite medium for which dielectric permittivity determines the Fresnel coefficients. Consequently, the intensity of the reflectivity lines is given by the values of the complex refractive index at the excitonic resonances. In bulk hBN, the situation is complex due to the spectral overlap of direct and indirect transitions above the bandgap. At present, there is no theoretical calculation of the dielectric permittivity of bulk hBN that takes into account the interplay between direct and indirect optical transitions in the 6-7 eV range. A simple consideration based on the direct exciton oscillator strength is oversimplified since it overlooks interference effects between the direct and indirect transitions in the exact value of the dielectric permittivity.

In summary, mBN on graphite and bulk hBN correspond to very different systems in terms of their reflectance properties so that no simple physical argument can be invoked to compare them, thus refuting point 4 of Reviewer 3.

Point 5 about "*the lack of a significant Stoke shift in the grown sample*"

Reviewer 3 writes "*The lack of a significant Stoke shift in the grown sample compared to the bulk sample does not support the crossover.*".

We are surprised by this comment in view of Fig.3a where we show that there is at least a ~200 meV detuning between the PL signal of mBN (around 6.1 eV) and the one of bulk hBN (extending below 5.9 eV).

Point 6 about "*the resonant excitation*"

Reviewer 3 raises several points dealing with our resonant excitation scheme, the first one being "*it is also unclear to me what I am supposed to see with the arrows in Fig. 3b.*"

As explained in the caption of Fig.3 (page 9), "*The vertical arrows indicate the Raman-shifted energy $E_{ex} - 2\Delta$, with $\Delta=156$ meV, corresponding to the energy of the LA(M) phonon in mBN.*" and in the main text, we wrote (page 9) "*We also note a distortion of the emission spectrum when varying E_{ex} , arising from the intrusion of resonant Raman scattering, 312 meV below the excitation energy [vertical arrows in Fig.3(b)], and superimposed upon the PL doublet (for more details, see*

Supplementary Section D)." Section D of our Supplementary Material provides a detailed description of the Resonant Raman Scattering (RRS) signal which appears at the energies indicated by the vertical arrows.

Reviewer 3 further criticizes that "*Also, there are too few excitation wavelengths performed to support the picture of resonant excitations.*" We do reject this criticism. We display four configurations of excitation to illustrate the resonant excitation scheme: two are at the energies of the PL doublet lines, one is in between, and the last one is out of resonance. A detailed discussion of these four cases is presented in Section D of our Supplementary Material, from which we provide a comprehensive picture of resonant excitation in mBN by means of femtosecond pulses in the deep UV.

LIST OF CHANGES

We provide below the list of the changes in our revised manuscript.

All changes in the revised manuscript are highlighted in red.

Change C1

In response to the Main point of Reviewer 1, we have added "or as trion absorption" in page 5. We have further changed the title of Section B of the Supplementary Material (from "BRIGHTENING OF TRIPLET DARK EXCITON" to "ORIGIN OF THE SECONDARY MINIMUM AT 6 eV IN REFLECTANCE SPECTRUM") and modified Section B in order to include the alternative scenario of trion absorption for the interpretation of the secondary minimum at 6 eV. In particular, we have added the following paragraph in Section B of the Supplementary Material:

"B2. Trion absorption

Although the first scenario provides a consistent picture of Fig.S4, one cannot exclude alternative interpretations, in particular trion absorption being the origin of the secondary minimum at 6 eV in the reflectance spectrum.

A trion is a three-particle bound state consisting of two electrons and a hole, or one electron and two holes. The contribution of trions in the optical response of semiconductors is well documented in the literature. Assuming that the ratio of trion to exciton binding energies in mBN is similar to the value of 10 % found in monolayer transition metal dichalcogenides [Mak et al. Nat. Mater. 12, 207 (2012)], one could also tentatively interpret the redshifted peak in reflectance in Fig.S4 as the trion absorption. By varying the sample growth conditions, the different intensities of the trion peak in Fig.S4 may be due to different levels of growth-induced residual doping, which are known to play a key role in transition metal dichalcogenides [Mak et al. Nat. Mater. 12, 207 (2012)] and also in carbon nanotubes [Matsunaga et al. Phys. Rev. Lett. 106, 037404 (2011)]."

Change C2

In response to the Main point of Reviewer 1, and in the continuity of Change C1, we have commented the peak multiplicity in the PL spectrum and discussed other possible scenarios in addition to the intervalley scattering and the momentum-dark excitons. Namely, in page 8, we have added the following sentences:

"Finally, we note the complexity of the PL spectrum goes beyond a simple doublet with additional peaks of lower intensity than the dominant doublet. While the 5.98 eV component may arise from the brightened triplet or the trion (see Supplementary Section B), the identification of the weak shoulders on the high-energy side will require more experimental and theoretical work in order to obtain a comprehensive understanding of the full emission spectrum."

Change C3

In response to Minor point 2 of Reviewer 1, we have added the (i) to (iv) labels in Fig.3b, for the four configurations of our resonant excitation experiments.

Change C4

In response to the Suggestion of Reviewer 2, we have added "*free-standing*" before mBN in page 9, and we have added "*Although the graphite substrate enhances screening in our epitaxial mBN,...*" in page 10.

Change C5

In response to Point 2 of Reviewer 3, we have added the following sentence in page 7:

"In contrast to transition metal dichalcogenides [3] we note the PL signal intensity of the indirect-gap bulk hBN is more intense than in the direct-gap mBN due to the unusually high internal quantum efficiency in bulk hBN [5,39]".

In relation to this change, we have added a new reference, which is Ref.39 of the revised manuscript.

[39] Schué, L., Sponza, L., Plaud, A., Bensalah, H., Watanabe, K., Taniguchi, T., Ducastelle, F., Loiseau, A. & Barjon, J. Bright luminescence from indirect and strongly bound excitons in h-BN. *Phys. Rev. Lett.* **122**, 067401 (2019).

Reviewers' comments:

Reviewer #1 (Remarks to the Author):

With the changes made to the manuscript I fully support its publication in Nature Communications

Reviewer #3 (Remarks to the Author):

As before, I do not see direct experimental evidence of monolayer BN as a direct band gap material. The authors did not present new experimental evidence to support the claim. The ways to show that a material is direct band gap in nature are standard: 1) measurement of the optical conductivity to show a sharp threshold increase in optical conductivity with photon energy by orders of magnitude. This is in contrast to an indirect gap material which typically shows a gradual increase in optical conductivity with photon energy. This is the most accepted method; 2) show significant enhancement in the PL quantum yield. This is less direct but if the amount of enhancement is significant (i.e. orders of magnitude), it is acceptable; 3) direct band structure measurement by ARPES but presumably not doable on BN. None of these has been performed in the study. Therefore I cannot be convinced as a matter of principle. In my opinion, the paper in its present form will just confuse people in the community.

Reviewer #3 (Remarks to the Author):

As before, I do not see direct experimental evidence of monolayer BN as a direct band gap material. The authors did not present new experimental evidence to support the claim. The ways to show that a material is direct band gap in nature are standard: 1) measurement of the optical conductivity to show a sharp threshold increase in optical conductivity with photon energy by orders of magnitude. This is in contrast to an indirect gap material which typically shows a gradual increase in optical conductivity with photon energy. This is the most accepted method.

Absorption (or “optical conductivity”, using Reviewer 3 terminology) in bulk hBN is very different from that in any other indirect gap semiconductor. The absorption spectrum does not show broad bands (“a gradual increase in optical conductivity with photon energy.”) but narrow lines, very much like that in direct semiconductors. This key point was first explained in our paper [Cassabois et al. Nat. Phot. 10, 262-266 (2016)]. It arises from the peculiar band structure of hBN. It leads to the involvement of phonons with a finite group velocity in the recombination processes, which itself results in a strong energy-dependence of the exciton-phonon matrix element. This technical point explains the origin of the narrow lines in the optical response of bulk hBN. It also gives rise to very sharp absorption edges (as evident from our two-photon excitation spectroscopy reported in Nat. Phot. 10, 262-266 (2016)), very much like that in direct gap semiconductors.

Thus, inspection of absorption edges in indirect-gap bulk hBN vs. direct-gap monolayer hBN is irrelevant to the demonstration of the direct band-gap crossover in the monolayer limit, not to mention the fact that the transitions occur at the same energy in both cases, as discussed in detail in our paper.

2) show significant enhancement in the PL quantum yield. This is less direct but if the amount of enhancement is significant (i.e. orders of magnitude), it is acceptable.

Indirect-gap bulk hBN is a very bright emitter, very much like direct gap semiconductors. This specificity was first pointed out in the pioneering work of Watanabe and Taniguchi on highquality hBN crystals [Nat. Mater. 3, 404 (2004)] and it led to the misinterpretation of bulk hBN as a direct-gap material until the publication of our paper in Nature Photonics. Recently, this qualitative description was complemented by quantitative estimations of the quantum yield, of the order of 40-50 % in bulk hBN, a remarkably large value for an indirect-gap material [Schué et al. PRL 122, 067401 (2019)].

Therefore, Reviewer’s 3 comparison of the quantum yield in indirect-gap bulk hBN vs. direct-gap monolayer hBN is not appropriate and is therefore irrelevant, as is the case for the optical absorption edges.

3) direct band structure measurement by ARPES but presumably not doable on BN. None of these has been performed in the study. Therefore I cannot be convinced as a matter of principle. In my opinion, the paper in its present form will just confuse people in the community.

ARPES provides access only to valence bands, and not to conduction bands so that ARPES experiments cannot demonstrate whether the gap is direct or indirect.

Moreover, we did report ARPES measurements, as explained on page 3 in our paper where we refer to paper [33] of our bibliography. We stress that in Ref.[33], we concluded that « the location of this valence band maximum [is] at the K point, in contrast to multilayer h-BN ». This was a first indication of the direct band-gap crossover, which is now demonstrated in our submitted manuscript by reflectance and photoluminescence spectroscopy and by selective optical pumping through resonant excitation of phonons.